# Melissopalynological Analysis of Honey from French Guiana

**DOI:** 10.3390/foods13071073

**Published:** 2024-03-31

**Authors:** Weiwen Jiang, Marie-José Battesti, Yin Yang, Élodie Jean-Marie, Jean Costa, Didier Béreau, Julien Paolini, Jean-Charles Robinson

**Affiliations:** 1Laboratoire COVAPAM, UMR QualiSud, Département Science et Technologies, Université de Guyane, 97300 Cayenne, French Guiana, France; wein.jiang@univ-guyane.fr (W.J.); elodie.jean-marie@univ-guyane.fr (É.J.-M.); didier.bereau@univ-guyane.fr (D.B.); 2UMR 6134 CNRS “Science pour l’environnement”, Faculté des Sciences et Techniques, Université de Corse, 20250 Corté, Corse, France; majobattesti@yahoo.fr (M.-J.B.); yin.yang@odarc.fr (Y.Y.); costa_d@univ-corse.fr (J.C.); paolini2b@gmail.com (J.P.); 3Office du Développement Agricole et Rural de Corse, 20600 Bastia, Corse, France

**Keywords:** *Apis mellifera*, honey, melissopalynology, pollen, flora, Tropical, Guiana Shield

## Abstract

Beekeeping directly depends on the floral biodiversity available to honey bees. In tropical regions, where nectar and pollen resources are numerous, the botanical origin of some honey is still under discussion. A precise knowledge of plants foraged by honey bees is useful to understand and certify the botanical origin of honey. In this study, attention was paid to honey samples from the French Guiana Atlantic coast where beekeepers generally place their hives in four types of biotopes: seaside vegetation, mangrove, savannah, and secondary forest. Pollen analysis of 87 honey samples enabled the identification of major plants visited by Africanized honey bees during the dry season (approximately from July to January). Through melissopalynologic analysis, 51 pollen types were identified and classified according to their relative presence. Frequently observed pollens (with relative presence > 50%) in French Guiana kinds of honey were those from *Mimosa pudica*, *Cocos* sp., *Rhyncospora* sp., *Avicennia germinans*, *Paspalum* sp., *Spermacoce verticillata*, *Tapirira guianensis*, *Cecropia* sp., Myrtaceae sp., *Mauritia flexuosa* sp., *Solanum* sp., and *Protium* sp. In many honeys, only *M. pudica* was over-represented (relative frequency > 90%). Color and electrical conductivity in French Guiana honeys exhibit significant variations, with color ranging from 27 mm to 110 mm Pfund, and electrical conductivity ranging from 0.35 to 1.22 mS/cm.

## 1. Introduction

French Guiana is the largest French department, with more than 90% of its territory (83,500 km^2^) still in preservation. In this region, the beekeeping season lasts about 9 to 10 months and matches the dry season (from May–June to December–January). Another period called “little summer of March” is also favorable to honey production. Although the territory is widespread, most beekeeping activity is mainly focused on the Atlantic coast. This French Guiana coastal sector includes lowlands and the northern shelf and can be divided into two distinct parts. The western part (from Rémire-Montjoly to Saint-Laurent-du-Maroni) is characterized as a “dry” zone, with an annual rainfall of between 1800 and 2500 mm. The eastern part (from Matoury to Saint-Georges) is identified as a “wet” zone, with annual rainfall of between 2500 and 3500 mm. The biome as a whole includes various biotopes, such as mangroves, savannahs, and tropical forests, all of which are visited by Africanized honey bees [1,2,3,4,5]. These climatic characteristics mean that beekeeping in French Guiana has developed mainly in the western part. However, apicultural production rarely exceeds 10 tons per year and is entirely sold in French Guiana under trade names such as “Amazonian forest honey”, “savannah honey”, or “seaside honey”. Despite this beekeeping commitment and the growing interest in this agricultural sector, the literature is poor regarding French Guiana bee-flower ecology [2,3,6,7,8,9].

Some palynological studies dealing with melliferous resources highlighted mostly *Mimosa pudica* L. flowers in *A. mellifera* foraging, leading to its prominence in French Guiana honeys [3,8]. *Avicennia germinans*, *Hyptis atrorubens,* and some palm species were also described as important nectar sources [2,3,8,10].

With a favorable climate, and nectariferous and polliniferous flowers from its Amazonian biome, French Guiana can consider reasonable economic outlooks with apiculture. From this perspective, enhancing our understanding of the flora visited by honey bees is essential to support and develop the French Guiana beekeeping sector.

For this purpose, the conventional method is melissopalynology, which consists of identifying pollen by microscopic analysis to determine the plants visited by honey bees during honey production [11,12]. This approach requires long examination times, the availability of a complete collection of pollen grains, and experts with experience in identification. However, it is the reference method for establishing the botanical origin of honey samples [13].

Quantifying pollen grain in relative frequencies (RF) across various honey samples from a given production region facilitates the establishment of connections between the pollen content (palynological characteristics) of honey and its corresponding biogeographical, ecological, and beekeeping contexts [11,12]. In French Guiana, studies employing a pollen counting method established by Louveaux and Maurizio [11] for *Apis mellifera* honeys are scarce. To our knowledge, only one such study has been conducted in this territory [9].

The aim of the present study included (1) the determination of the pollen spectrum present in the French Guiana honeys and (2) enlisting nectariferous and polliniferous plants visited by honey bees. Through these two research axes, it would be feasible to establish a list of plants exhibiting significant importance for apiculture in this region. These data may also serve as the foundation for future research on the interactions between plants and bees.

## 2. Materials and Methods

### 2.1. Honey Sampling

Eighty-seven ripened types of honey (H1 to H87) were harvested during the dry season (from July to December/January). They were centrifuged with combs and directly picked up from the maturer’s exit before storage at 20 °C. These samples were collected from seven localities in French Guiana (Awala-Yalimapo, Saint-Laurent-du-Maroni, Sinnamary, Kourou, Macouria, Montsinéry-Tonnegrande, and Rémire-Montjoly) during three different sampling periods (from August 2014 to January 2015: samples H1–H17; from July 2015 to January 2016: samples H18–H61; and from September 2016 to May 2017: H62–H87). Apiaries were mainly located in the central and western French Guiana coastline (Figure 1). The harvesting sites are distributed as follows: Awala-Yalimapo has one site numbered 1, Saint-Laurent-du-Maroni has two sites numbered 2 and 3, Sinnamary has four sites numbered 4, 5, 6, and 7, Kourou/Macouria has four sites numbered 8, 9, 10, and 11, Montsinéry-Tonnegrande has four sites numbered 12, 13, 14, and 15, and Rémire-Montjoly has one site numbered 16.

### 2.2. Melissopalynological Analysis

To create a pollen bank, all species in bloom within a radius of 2 km around each apiary were sampled for 3 years (2014–2017). Except for large trees (> 20 m), pollen from all accessible flowers was sampled. For each species, pollen was collected, kept in a hydro-alcoholic solution (10:90-*v*/*v*), and stored at 20 °C until further use. After centrifugation, sediments (fresh pollen) obtained were fully transferred and spread out with a micropipette on a slide located on a hot plate at 40 °C. Fresh pollen was degreased with diethyl ether (Fisher Scientific, France), embedded in Kaiser’s glycerol gelatin (Merck, France), and colored by Ziehl Fushin (Reactif RAL, France). Varnish was applied on each coverslip to protect pollen and the slide against moisture. These pollen slide bases were set up as our reference library.

Pollen extraction from honey was carried out according to the method described by Louveaux et al. [11] and Von der Ohe et al. [12] and adapted by Battesti et al. [14,15]. Approximately 10 g of honey was dissolved in 40 mL of acidic water (0.5%-*v*/*v*). The mixture was stirred until complete dissolution and centrifuged. The liquid phase was eliminated, and sediment was washed twice with distilled water before the second centrifugation. Finally, sediment was entirely transferred and spread out with a micropipette on a slide. Sediment was degreased and embedded in Kaiser’s glycerol gelatin, and slides were coated with varnish.

The pollens from honey slides were analyzed, identified, and counted following the method described by Louveaux et al. [11] and adapted by Battesti et al. [14].

First, all taxa were identified by quick scan with an ocular (magnification ×50 and ×100) microscope. Pollen identification was carried out by comparing reference library slides (Figure 2) and/or literature data (PalDat and Oreme databases). Observation of fungal spores, soot particles, fine granular mass, and mineral particles were also included in the analysis. Honey bees usually collect these components during the harvesting of honeydew and are often considered as indicators of honeydew contributions [16]. Then, each taxa distribution was estimated by counting (magnification ×50) with a microscope’s field of view in order to value (i) the total pollen spectrum (qualitative analysis) of each honey sample, which was expressed in terms of pollen relative frequency (RF), and (ii) pollen density (quantitative analysis), which was qualified by the absolute number of pollen grains in 10 g of honey (PG/10 g). It was necessary to produce a pollen spectrum for all honey samples to determine the relative presence (percentage of taxon’s presence across the 87 samples) of each taxon. As for the relative frequency, it was calculated for each taxon in each sample by dividing the number of counted grains for a specific taxon by the total number of counted grains in the sample.

Pollen analysis from the honey slide involves examining several gridded counting fields across multiple lines, with a minimum of three lines used to cover the top, center, and bottom of the slide. These individual counting fields were evenly distributed along each counting line to ensure a uniform examination of the slide. The specific number of fields per line was determined by the observed pollen density. Pollen density was calculated using the following formula [12]:PG/10 g = Ng/Nc × NTC × 10/Pm

Ng is the total number of counted pollen grains; Nc is the total number of studied microscopic fields; NTC is the total number of microscopic fields (obtained by dividing the total surface area of the deposit by the surface area of a microscopic field at the relevant magnification), and Pm is the mass of analyzed honey.

Honeydew indicators (fungal spores, mycelium, and algae cells) were labeled according to frequency of observation. It was rare (R) when indicators of honeydew were present in one out of every two fields; frequent (F) when they were present in all fields of observation without reaching a high number (less than 10); and very frequent (VF) when presence was greater than 10 at each counting field. Attention was also given to the presence of fine granular mass and the presence of mineral particles that were elements that could indicate honeydew presence [16].

Parameters such as minimum, maximum, relative presence, standard deviation (SD), and relative standard deviation (RSD) were rigorously calculated to provide an overview of all the melissopalynological data.

### 2.3. Physico-Chemical Test

To complete characterization, electrical conductivity and water content were assessed. Electrical conductivity was measured using a micro CM2210 (CRISON, Barcelona, Spain) conductivity meter at 20 °C, and water content (moisture) was determined by refractometric method with a PAL-22S (Atago^®^, Tokyo, Japan) refractometer at room temperature using the Bogdanov method [17]. The physicochemical tests for honey samples were carried out during their reception.

## 3. Results

### 3.1. Relative Presence Analysis of Pollen Grains

Sixty-nine pollen types were listed with analyses of honey slides (H1 to H87) (Appendix A). About half of them were present in more than 10% of samples. Using the pollens’ bank and the literature, 51 taxa were identified (from 36 families), but 18 remained unknown and were noted as NI 1–18 (Appendix A). Pollens of *Coco* sp., *Elaeis* sp., *Xyris* sp., *Ilex guianensis*, *Merremia* sp., Chenopodiaceae /Amaranthaceae, *Ceiba pentandra*, Ranunculaceae, *Desmanthus* sp., Urticaceae, Sapindaceae, *Serjania* sp., and *Citrus* sp. have been identified through the literature [18,19,20,21,22,23,24,25,26,27,28,29,30].

*M. pudica* (Fabaceae) was the only pollen type observed in all samples. Eleven other taxa were present in more than 50% of samples (Table 1 and Appendix A), these include *Cocos* sp. (Arecaceae) (98.9%), *Rynchospora* sp. (Cyperaceae) (89.7%), *A. germinans* (Acanthaceae) (88.5%), *Paspalum* sp. (Poaceae) (88.5%), *Spermacoce verticillata* (Rubiaceae) (83.91%), *Tapirira guianensis* (Anacardiaceae) (81.6%), *Cecropia* sp. (Urticaceae) (79.3%), *Myrtaceae* (75.9%), *Mauritia flexuosa* (Arecaceae) (70.1%), *Solanum* sp. (Solanaceae) (64.4%), and *Protium* sp. (Burseraceae) (58.6%). These twelve pollen types were present in more than 50% of the samples, and they highlighted the most typical botanical families of the French Guiana beekeeping repertory. They are qualified as « regional constancy ». Twenty-two pollen types were observed between 10 and 50%, and they were considered as part of regional diversity. Lastly, thirty-five taxa were present in less than 10% of honeys. Six unidentified pollens were present in more than 20% (Appendix A).

### 3.2. Relative Frequency (RF) Analysis of Pollen Grains

Pollen types were correlated to samples according to their RF (relative frequency) (Table 1 and Appendix A), taxa that are indicative of French Guiana beekeeping inventory are those that have a high relative presence with high RF (greater than 45%). In this study, only *M. pudica* has this specificity. Indeed, it was present in all samples and had an RF at least once above 45% in 63 samples. In addition, its relative standard deviation (RSD) was very low due to small values dispersion around the mean. Furthermore, Appendix A illustrated that *M. pudica* was often over-represented with RF greater than 90% in 33 pollen spectra and less than 16% for only 7 pollen spectra. These statistical data showed that *M. pudica* RF was regularly above 45%. The data confirmed that this species has an important place in the French Guiana beekeeping landscape. *T. guianensis* and *Protium* sp. also had RF at least once greater than 45% but the average value of their RF was very low (<10%), with high RSD (Table 1 and Appendix A). Statistical data showed that both species, although occurring in numerous honeys, had high RF in a few samples. In conclusion, *M. pudica*, *T. guianensis,* and *Protium* sp. belonged to the dominant pollen group but only *M. pudica* had RF > 45% in several samples, in contrast to *T. guianensis* and the genus *Protium* sp.

### 3.3. Pollen Typology of Honeys from French Guiana

On presence/absence (relative presence) and relative frequency basis, a first typology of French Guiana honeys was suggested, which was classified into two groups. The first group included samples with a dominant pollen (RF > 45%). Sixty-three kinds of honey had *M. pudica* as “dominant” pollen, five kinds of honey where *T. guianensis* was major (H37, H44, H46, H48, H49), and one honey where *Protium* sp. RF was greater than 45% (H47) were concerned (Appendix A). The second group included 18 remaining samples (H5, H9, H11, H13, H14, H34, H35, H45, H58, H61, H62, H68, H78–82, H85), which had a more complex pollen spectrum without dominant taxa.

### 3.4. French Guiana Honey Profile

The majority of samples from French Guiana belonged to honey class II or III (Appendix A), these data suggest that the nectar may have been sourced from plants with either a normal pollen type (the amount of nectar used in honey production is proportional to the amount of pollen grains) or an overrepresented pollen type (the amount of pollen grains is not proportional to the quantity of nectar collected by honey bees). Two samples (H4, H44) had a pollen density higher than 1.000.000 grains per 10 g of honey.

Statistical data of physico-chemical parameters per area were reported in Table 2. Except for a few samples, the moisture content of honeys is always below 20%. For parameters such as color and electrical conductivity, French Guiana honeys have relatively large variations. The color varies from 27 mm to 110 mm Pfund. Averages and standard deviations indicate that French Guiana honeys have a color varying between very light amber (34 to 50 mm Pfund) and light amber (50 to 85 mm Pfund). The electrical conductivity ranges from 0.35 to 1.22 mS.cm^−1^. Physico-chemical values were included in Appendix A to provide additional information on the samples.

Classification of honey samples by harvesting zone reveals the following characteristics. In the Awala-Yalimapo area, taxa at least once dominant (RF> 45%) were as follows: *M. pudica*, *T. guianensis,* and *Protium* sp. The samples H34, H35, H62, and H85 were untypical because they did not have dominant pollen but two secondary pollen types (RF> 16%), which were *M. pudica* and *T. guianensis*; it was the same case for H85, with *A. germinans* and *T. guianensis*. For non-identified pollen, only NI 2 had RF > 16%. In this sector, H35 and H47 had electrical conductivity higher than 0.80 mS.cm^−1^ but had light colors and did not have significant honeydew indicators (Table 3 and Appendix A).

In the Saint-Laurent-du-Maroni region, seven out of ten harvested honeys featured a strong presence of *M. pudica*. Only H13, H45, and H46 had an RF lower than 45%. Other taxa that reached at least once RF > 10% were as follows: *T. guianensis*, *Cecropia* sp., *Diplotropis purpurea*, Asteraceae, and Scrophulariaceae (Table 4 and Appendix A). H67 had an electrical conductivity higher than 0.80 mS.cm^−1^, a significant honeydew indicator, and a pronounced amber color. Honeydew may have played a role in the production of H67 (Table 4).

The highest RF for *Cecropia* sp. was observed in honeys from Awala-Yalimapo and Saint-Laurent-du-Maroni; and except for these both areas, *Cecropia* sp. RF was always lower than 3%.

Sinnamary beekeepers defined two harvesting zones. Honeys from hives on the seaside (sites 4 and 5—Figure 1) are called seaside honeys, while southern harvests are called Amazonian forest honey (sites 6 and 7–Figure 1). We noted that *A. germinans* pollen was more important in seaside honeys. (Table 5 and Appendix A). Thus, H78 and H81 were the two honeys with the highest RF for *A. germinans* (43.9% and 41.6%, respectively). For Amazonian forest honeys, *M. pudica* was dominant in sixteen samples. Only three honeys had *M. pudica* less than 45%. Other taxa with RF greater than 10% were as follows: *M. flexuosa, Rynchospora* sp., *A. germinans, S. verticillata, Emmotum fagifolium,* and NI5 (Table 6 and Appendix A). In the case of seaside honeys, 13 out of 19 samples were dominated by *M. pudica* pollen (from 55.2% to 98.7%). In six other kinds of honey, *M. pudica* and *A. germinans* had RF > 10% and oscillated between secondary and minor important pollen. The vast majority of Sinnamary honeys had electrical conductivity less than 0.80 mS.cm^−1^. Only five samples (H22–H26), which were harvested during the year 2015, possessed electrical conductivity higher than 0.80 mS.cm^−1^; but, honeydew indicators are rare or absent (Table 5 and Table 6).

A Beekeeper who has beehives in sites 8, 9, and 10 (Kourou/Macouria—Table 7 and Appendix A) mixed honey from these different sites. Whereas samples from site 11 were produced by a single apiary. *M. pudica* was dominant in five types of honey (H36, H38, H42, H43, and H73). As for H37 and H44, *T. guianensis* (50 and 83.4%, respectively) was major. Only H36 had electrical conductivity greater than 0.80 mS.cm^−1^, with rare honeydew indicators.

Honeys of Montsinéry-Tonnégrande came from two sectors. The first sector was close to the sea and included sites 12 and 13. Sites 14 and 15 were away from the coast (Figure 1). For sites 12 and 13, seven out of eight honeys were dominated by *M. pudica* with RF between 47% and 98.6%. In this sector, other taxa with RF at least once greater than 10% were as follows: *Cocos* sp., *M. flexuosa, T. guianensis,* and *Myrtaceae* sp (Table 8 and Appendix A). For sites 14 and 15, eighteen pollen forms were identified following analysis. *M. pudica* was the dominant pollen in all honey with RF between 58.3% and 98.6%. Except for *M. pudica*, taxa with RF > 10% were *T. guianensis* (14.8% and 10.9% in H51 and H59, respectively) and *Spondias mombin* (11.1% in H59). Harvested honey in site 14 always had high electrical conductivity. Only H39 had a significant honeydew indicator, but its color was rather light (55 mm Pfund). These data suggest that the high electrical conductivity may be caused by a different element than honeydew.

Finally, the honeys from Rémire-Montjoly have a pollinic profile marked by the presence of *S. mombin* (RF always greater than 15%). *M. pudica* kept high RF but was dominant only in H10 and H41. All samples harvested in this area had electrical conductivity less than 0.80 mS.cm^−1^ (Table 9 and Appendix A).

A comparative analysis of pollen profiles from different harvest areas (Figure 3, Table 10) allows for a more precise observation of the micro-regional trend discussed earlier. Thus, honeys from the western part of French Guiana (Awala-Yalimapo and Saint-Laurent-du-Maroni) are characterized by a higher concentration of *T. guianensis*, *Cecropia* sp., *Protium* sp., and Scrophulariaceae sp. Sinnamary has honeys rich in *A. germinans* pollen, while the Macouria/Kourou area is distinguished by a combined richness in *T. guianensis* and *Solanum* sp. The presence of *S. mombin* is significantly higher in Rémire-Montjoly, and Montsinéry-Tonnégrande is characterized by honeys in which the main taxa are poorly represented.

## 4. Discussion

### 4.1. Contribution to the Selection of Botanical Taxa Usable as Geographic Markers

A literature review was conducted on the tropical taxa found in our study to identify potential geographic markers. Among the identified taxa, twenty-nine taxa are tropical: *Cocos* sp., *A. germinans*, *Avicennia* sp., *S. verticillata*, *T. guianensis*, *Cecropia* sp., *M. flexuosa*, *Protium* sp., *S. mombin*, *D. purpurea*, *Elaeis* sp., *Piper marginatum*, *E. fagifolium*, *Mimosa pigra*, *Miconia* sp., *H. atrorubens*, *Davilla rugosa*, *Myrcia tomentosa*/*Myrcia sylvatica*, *C. pentandra*, *Vismia guianensis*, *Vismia latifolia*, *Vochysia* sp., *Dalbergia ecastaphyllum*, *Rolandra ruticose*, *Couroupita guianensis*, *Inga* sp., *Anacardium occidentale*, *Pachira aquatica,* and *Serjania* sp. [2,31,32,33,34,35,36,37,38,39,40,41,42,43,44,45,46].

Among them, *T. guianensis*, *D. rugosa,* and *Vochysia* sp. were rare or absent in the Caribbean area. They are, however, present in Central America, the Guiana Shield region, and the Amazon basin [33,37,39,42,44,46,47,48]

Samples containing *Ilex guianensis* pollen can be located in a production area between southern Mexico and the Guiana Shield [46]. Although the biogeographic distribution of *Ilex guianensis* includes French Guiana, its distribution was too broad and not specific enough to be used solely as a criterion for discriminating the geographic origin of honeys from French Guiana.

*M. flexuosa*, *D. purpurea*, *E. fagifolium,* and *V. guyanensis* possess a distribution area less wide than *Ilex guianensis*; their presence in honeys allows the location of harvest area where the Guiana shield and Amazon basin are included. *V. latifolia* has an interesting geographic distribution. Indeed, it is confined to the region of the Guianas Shield region [32,35,39,40,49,50,51,52,53]. Honeys containing its pollen are highly likely to originate from a geographic area delineated by the Guianas Shield.

Even though it is not specific to French Guiana, these nine taxa mentioned above allow us to geographically restrict the area of sample collection between Central America (e.g., *Ilex guianensis*) and the Amazon basin (e.g., *M. flexuosa*), excluding the Caribbean area. *V. latifolia* is the taxa that allows for geographically locating honey collected in the Guiana Shield. In the context of French Guiana, where the flora is very similar to neighboring regions, the selection of these nine taxa was a first step towards French Guiana honey’s geographical origin characterization. In addition, they were permitted to affirm that samples were indeed produced in the tropical zone in a geographical area including French Guiana.

The combination of these nine taxa in samples and RF analysis will permit future studies to indicate the harvest zone. For instance, it is possible that French Guiana honeys do not possess the same RF in *M. flexuosa* pollen grains as honeys from neighboring countries.

*M. pudica* may also be included in this list. Its pollen is already reported with high RF (> 45%) in honeys from Brazil, Mexico, and China. However, values are always remaining below 90% [54,55,56,57,58]. According to Roubik’s works, the high dominance of *M. pudica* is not surprising. This taxon has great bioavailability in French Guiana and is widely visited by *A. mellifera* for pollen. *M. pudica* pollen can constitute 89% of the total pollen pellets [3,8]. This high occurrence of *M. pudica* pollen in French Guiana honeys may stem from secondary contamination, including pollen transported by the wind or linked to honey bee activity. To our knowledge, no publication mentioned *M. pudica* with RF greater than 90%.

Other species of the genus can have an RF of over 90%. This is the case of *M. tenuiflora* in the *Melipona* honeys harvested in the Caatinga region of Brazil [59]. Its high pollen presence (> 90%) in some honey may contribute to its geographical origin marking. Analyses conducted by Kerkvliet [26] showed that *M. pudica* is present at RF between 1 and 30% (average of 11,8%) in Surinam honeys. Thus, RF reached by *M. pudica* (RF> 90%) in some honeys and its association with other taxa could also potentially be considered a French Guiana specificity in the Guiana Shield area.

Taken together, no species endemic to French Guiana have been found in our honey samples. However, it is possible to propose a delimitation for the geographical origin of analyzed samples by cumulating several species (*T. guianensis*, *D. purpurea*, *E. fagifolium*, *M. flexuosa*, *D. rugosa*, *V. latifolia*, *V. guianensis*, genus *Vochysia* sp., and *Ilex guianensis*).

### 4.2. Contribution to Understanding Botanical Origins of French Guiana Honeys

Nectar constituting honey can be produced from the flower (nectar honey), from secretions located out of the flower (other living parts), or from the excretion of phytophagous insects (honeydew honey) [60]. Based on the list of melliferous plants’ pollens found in French Guiana honeys (Appendix A), this study tried to highlight taxa that could have nectariferous interest and/or could shelter honeydew-producing insects.

Firstly, taxa with nectaries are distinguished from those without. Nine taxa were devoid of floral and extrafloral nectaries: *Rhynchospora* sp., *M. flexuosa*, *Elaeis* sp., *Xyris* sp., Urticaceae sp., major species of *Paspalum* genus, *P. marginatum*, major species of *Miconia* genus, and some species of Chenopodiaceae/Amaranthaceae family [61,62,63,64,65,66]. *M. pudica* and *M. pigra* can also be included in this list. Indeed, these two species belong to a genus botanically subdivided into five sections (Mimadenia, Batocaulon, Calothamnos, Habbassia, and Mimosa) where each section has a relatively large number of species. However, only species belonging to the Mimadenia section have nectaries (foliar nectaries). However, *M. pudica* and *M. pigra* are classified in the Mimosa section [56,67]. Thus, according to botanical literature, *M. pudica* and *M. pigra* would not be nectariferous species [3,8,59,67,68]. Apart from these eleven taxa and except for non-identified species, other taxa in Appendix A would have nectar glands.

Another parameter is the beekeeping interest of certain taxa, which varies according to the sex of the plant. For instance, the *T. guianensis* male plant produces larger flowers that provide pollen and nectar, while female individuals have smaller flowers producing only nectar (anthers devoid of pollen) but have better longevity [69,70]. The same case is roughly found in *Protium* sp. [71]. Floral morphology must also be studied. Some taxa are known to be nectariferous, but their pollen is only present in small amounts in honey like *C. pentandra* (Bombaceae). In this species, most of these flowers are bowing downwards, which often leads to nectar drainage. In addition, the length of the stamens allows foragers to harvest precious liquid without coming into contact with anthers [72]. This would explain the low RF (< 3%) of this taxon in honeys from French Guiana.

Data on the percentage of pollen brought by bees per harvested nectar (pollen type of botanical species) are absent in South America. These elements related to the floral structure are essential for determining the botanical origin of French Guiana honeys [12,73,74].

However, in this study, pollen grain content, which is an essential part of the characterization, was calculated for each honey sample. Each plant species has its pollen representation [14]. As a result, honeys from plants where a specific type of pollen is over-represented always have a high pollen content [11]. Chestnut honey (RF of chestnut pollen grains > 90%) is a notable example [75]. Conversely, honeys from species whose pollen types are under-represented systematically have a low pollen content [14]. For example, *Arbutus unedo* is recognized as having an under-represented pollen type due to the shape of its flower and the large size of its pollen [76].

According to Louveaux et al. [11], the pollen density of our samples allows us to put forward a hypothesis that French Guiana honeys can possibly come mainly from nectariferous resources having normal and/or over-represented pollen types.

In addition, there is also the question of taxa with extrafloral nectaries. In this study, there are eight: *A. germinans*, some species of *Cecropia* sp., *Solanum* sp., *Protium* sp., *Acacia mangium*, *Merremia* sp., *D. ecastaphyllum,* and *Inga* sp. The presence of extrafloral nectaries can vary in the Poaceae, Myrtaceae, and Asteraceae families [31,65,77,78,79,80,81,82,83,84,85]. However, limited research has explored the association between honey bees and extrafloral nectaries in tropical areas [7,86,87]. Quantifying their influence on tropical honey composition is a challenging endeavor.

Few samples had electrical conductivity higher than 0.80 mS.cm^−1^. These values may reflect honeydew presence [60,73]. However, this elevated electrical conductivity is not consistently accompanied by other indicative criteria of honeydew, such as a dark color or the presence of figurative elements reflecting honeydew presence. Conversely, some samples where electrical conductivity is low (< 0.80 mS.cm^−1^) present, in contrast, a higher proportion of honeydew indicator elements. This was the case of samples H10, H41 (Rémire-Montjoly), H17 (Montsinéry-Tonnegrande), H71, H77, H79, H80, H81, H82 (Sinnamary), and H85 (Awala-Yalimapo).

It should be noted that the honey bees can collect pollen opportunistically. For example, *A. mellifera* does not have the skills to extract pollen from the anthers of *Solanum* sp. flowers. It collects pollen left by other insects on extra-floral parts. In the process, it can also collect spores, hyphae, or algae. This would explain the presence of honeydew indicators for some samples [77,88].

These data could be indicative of differential behavior between tropical and European honeys facing the presence or absence of honeydew. It is possible that electrical conductivity does not seem to be a major factor in determining the type of honey (flower or honeydew) in tropical areas. Indeed, the work carried out in Colombia showed that only 35% of honeys with electrical conductivity greater than 0.80 mS.cm^−1^ have been confirmed as honeydew honeys by the high number of hyphae, spores, and *Quercus humboldtii* (oak tree) pollen found in it. The oak tree is the habitat of honeydew-producing *Stigmacoccus asper* [89].

In the current state of knowledge, it is laborious to draw definitive conclusions about the actual botanical origins of French Guiana honeys. Nevertheless, our data permit us to discern certain botanical tendencies, as seen in the case of honey H37, H44, H46, H48, and H49, which exhibit a high pollen content of *T. guianensis* grains. These honeys could be classified as multi-floral honeys with a *T. guianensis* tendency.

These results emphasized the remaining issues to be explored: (i) in addition to flower nectar, it is possible that bees may also collect extrafloral nectar and/or honeydew. The study of conditions that would attract honey bees towards these resources is in our perspective; (ii) the study of plant-bee relationships needs to be deepened to obtain data on various species pollen types, which are visited by foragers; (iii) the presence of non-identified pollen showed that our pollen bank must be enriched. Large trees (> 20 m) are likely great nectar providers. Their study and sampling of pollen from their flowers are essential to complete this work.

## 5. Conclusions

This melissopalynological work highlights the fact that *Africanized honey bees* visit very few taxa despite flora richness around apiaries. Eleven botanical families were considered as important for beekeeping in French Guiana (Mimosaceae, Asteraceae, Arecaceae, Cyperaceae, Verbanaceae, Poaceae, Rubiaceae, Anacardiaceae, Urticaceae, Solanaceae, and Burseraceae).

It appears that some harvesting areas had specific pollinic characteristics (e.g., Sinnamary stood out with a higher presence of *A. germinans*; Rémire-Montjoly is distinguished by *S. mombin* pollen).

Additional research is required to explore the diversity of nectar sources, aiming to more precisely ascertain the botanical origin of honeys from French Guiana.

## Figures and Tables

**Figure 1 foods-13-01073-f001:**
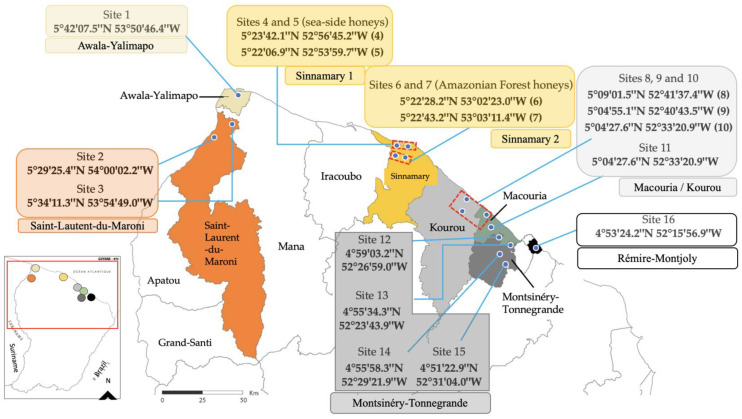
Localization of the sampling sites. Distribution of samples: Awala-Yalimapo (Site 1: H32, H34, H35, H47–49, H62, H85); Saint-Laurent-du-Maroni (Site 2: H45; Site 3: H1, H2, H7, H13, H33, H46, H67, H83, H84); Sinnamary (Sites 4 and 5: H3, H8, H14, H18–20, H22, H24, H25, H54, H56, H58, H60, H61, H74, H76, H78, H80, H81; Sites 6 and 7: H4, H9, H15, H21, H23, H26, H27, H52, H53, H55, H57, H69, H70–72, H75, H77, H79, H82); Macouria/Kourou (Sites 8, 9 and 10: H36–38, H73; Site 11: H42–44); Montsinéry-Tonnegrande (Site 12: H30; Site 13: H6, H11, H12, H16, H17, H29, H31; Site 14: H39, H40, H50, H51; H59, H63–H66, H86, H87; Site 15: H28); Rémire-Montjoly (Site 16: H5, H10, H41, H68).

**Figure 2 foods-13-01073-f002:**
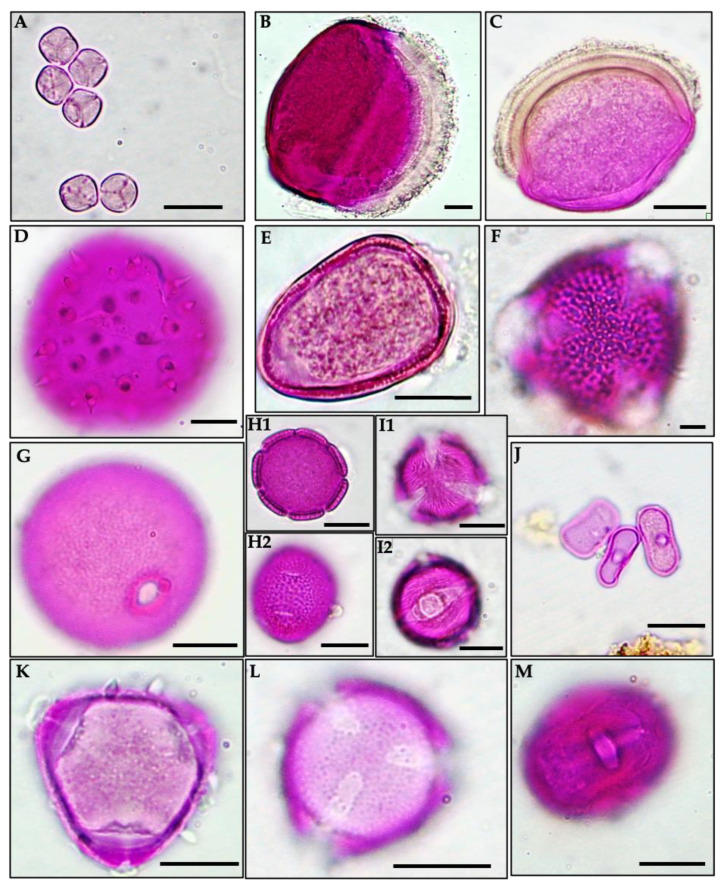
Examples of pollen present in the laboratory library slides. (**A**): *Mimosa pudica*; (**B**): *Cocos* sp.; (**C**): *Euterpe oleracea*; (**D**): *Mauritia flexuosa*; (**E**): *Rhynchospora cephalotes*; (**F**): *Avicennia germinans*; (**G**): *Paspalum maritum*; (**H1**,**H2**): *Spermacoce verticillata*; (**I1**,**I2**): *Tapirira guianensis*; (**J**): *Cecropia* sp.; (**K**): *Eucalyptus* sp.; (**L**): *Solanum leucocarpon*; (**M**): *Protium heptaphyllum*. Scale bars—10 μm.

**Figure 3 foods-13-01073-f003:**
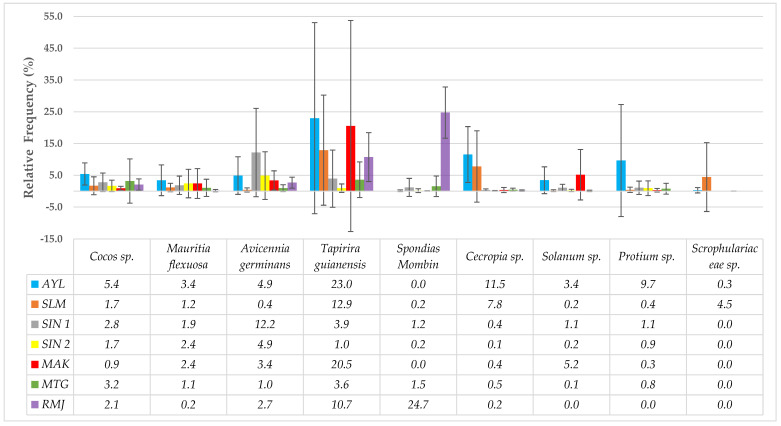
Comparative analysis of pollen profiles from different harvesting zones. AYL: Awala—Yalimapo; SLM: Saint—Laurent—du—Maroni; SIN 1: Sinnamary (seaside honeys); SIN 2: Sinnamary (Amazonian forest honeys); MAK: Macouria/Kourou; MTG: Montsinéry—Tonnégrande; RMJ: Rémire—Montjoly. The Focus was on taxa present in more than 50% of the samples and having an RF at least once greater than 16%. Each histogram represents the average RF of the taxon for a given site. For example, *Coco* sp. is present in eight samples from Awala—Yalimapo (AYL), with RF ranging from 0.3 to 9.0%. The mean RF was 5.4%, with an SD = 3.5%. *M. pudica* was not selected due to its excessively high values, overshadowing those of the other taxa.

**Table 1 foods-13-01073-t001:** Main pollen types identified in the French Guiana honeys (more than 50%).

Taxa		Relative Frequency								Presence in Samples		
		Less than 3%		Between 3% and 16%		Between 16% and 45%		Greater than 45%		Number Sample		Relative Presence (%)

*Mimosa pudica* L.		2b		1a, 2b, 1c, 1e		2a, 3b, 5c, 3d, 1e, 1f, 2g		7a, 1b, 13c, 16d, 5e, 19f, 2g		87		100.0
*Cocos* L.		8a, 2b, 12c, 15d, 7e, 14f, 3g		1a, 6b, 7c, 4d, 5f, 1g		1f		-		86		98.9
*Rhynchospora* Willd.		7a, 5b, 15c, 16d, 7e, 17f, 2g		1a, 1c, 2d, 3f, 2g		-		-		78		89.7
*Avicennia germinans* (L.) Stearn		6a, 2b, 7c, 10d, 3e, 16f, 3g		4b, 6c, 7d, 4e, 1f, 1g		6c, 1d,		-		77		88.5
*Paspalum* (L.) Trin.		5a, 7b, 16c, 19d, 6e, 18f, 4g		2c,		-		-		77		88.5
*Spermacoce verticillata* L.		7a, 4b, 17c, 17d, 6e, 18f, 1g		1c, 1d, 1f		-		-		73		83.9
*Tapirira guianensis* Aubl.		4a, 2b, 7c, 15d, 3e, 9f, 1g		3a, 1b, 4c, 2d, 1e, 6f, 2g		2a, 1b, 1c, 1f, 1g		1a, 2b, 2e		71		81.6
*Cecropia* Loefl.		4a, 2b, 17c, 13d, 3e, 16f, 2g		4a, 3b,		2a, 3b		-		69		79.3
Myrtaceae Juss.		5a, 7b, 14c, 11d, 4e, 17f, 4g		2c, 1d, 1f		-		-		66		75.9
*Mauritia flexuosa* L.f.		5a, 3b, 13c, 9d, 6e, 11f, 1g		1a, 3b, 3c, 3d, 1e, 1f		1d		-		61		70.1
*Solanum* L.		5a, 4b, 16c, 12d, 2e, 10f		3b, 2c, 1e		1e		-		56		64.4
*Protium* Burm.f.		3a, 3b, 10c, 12d, 2e, 12f		3c, 2d, 2f		1b		1b		51		58.6

Legend for “relative frequency” column 1a, 2b. The number represents the number of samples and the letter represents the sampling area as follows: a = honeys collected in the commune of Saint-laurent-du-Maroni; b = honeys collected in the commune of Awala-Yalimapo; c = honeys collected in the commune of Sinnamary (seaside apiary); d = honeys collected in the commune of Sinnamary (Amazonian forest apiary); e = honeys collected in the commune of Macouria and Kourou; f = honeys collected in the commune of Montsinéry-Tonnegrande; g: honeys collected in the commune of Rémire-Montojoly.

**Table 2 foods-13-01073-t002:** Physico-chemical data.

Locality	Moisture (%)			Color (mm Pfund)			Electrical Conductivity (mS/cm)			Number Sample
	min.	max.	mean ± SD	min.	max.	mean ± SD	min.	max.	mean ± SD	
**Awala Yalimapo** (site 1)	17.8	19.8	18.5 ± 0.7	35	99	59 ± 19,9	0.51	1.02	0.74 ± 0.16	**8**
**Saint-Laurent-du-Maroni** (sites 2 and 3)	17.8	21.9	19.6 ± 1.4	41	99	70.3 ± 17.3	0.48	0.93	0.78 ± 0.14	**10**
**Sinnamary (seaside honeys)** (sites 4 and 5)	16.6	19.1	17.6 ± 0.9	27	83	49.4 ± 16.7	0.44	0.93	0.62 ± 0.16	**19**
**Sinnamary (forest honeys)** (sites 6 and 7)	16.4	19.3	17.8 ± 0.9	27	110	70.5 ± 27.1	0.35	0.96	0.58 ± 0.15	**19**
**Macouria/Kourou** (sites 8, 9, 10 and 11)	18.2	21.4	19.4 ± 1.2	27	92	65.3 ± 23.5	0.47	0.88	0.67 ± 0.13	**7**
**Montsinéry-Tonnegrande** (sites 12, 13, 14 and 15)	15.8	21.0	18.4 ± 1.3	35	83	55.0 ± 10.4	0.52	1.22	0.86 ± 0.17	**8**
**Rémire-Montjoly** (site 16)	13.7	20.7	17.9 ± 2.9	46	92	69 ± 21.9	0.58	0.77	0.67 ± 0.87	**4**

Pfund color scale: <9 mm (water white); 9–17 mm (extra white); 18–34 mm (white); 35–50 mm (extra light amber); 51–85 mm (light amber); 86–114 mm (amber); >114 mm (dark amber).

**Table 3 foods-13-01073-t003:** Pollen spectra of honey collected in Awala-Yalimapo from French Guiana.

**Pollen Type** Present in more than 50% of samples with RF at least once greater than 10% found in the honey samples from French Guiana (across all sampling sites)	**Awala—Yalimapo**							
	**Site 1**							
	H32	H34	H35	H47	H48	H49	H85	H62
	Sept, 2015	Sept, 2015	Oct, 2015	Dec, 2015	Dec, 2015	Dec, 2015	Nov, 2016	Mar, 2017
**Mimosaceae**								
*Mimosa pudica*	94.0	39.4	27.8	2.9	1.1	6.3	9.1	20.4
**Arecaceae**								
*Cocos* sp.	0.6	9.0	5.9	5.3	4.8	0.3	8.4	9.0
*Mauritia flexuosa*	-	-	0.1	5.6	1.5	0.9	14.0	5.5
**Cyperaceae**								
*Rhynchospora* sp.	0.6	1.9	-	-	0.2	2.0	-	0.4
**Acanthaceae**								
*Avicennia germinans*	-	0.2	0.3	9.9	5.9	-	16.0	7.1
**Anacardiaceae**								
*Tapirira guianensis*	0.3	-	0.1	-	70.8	65.3	32.4	14.9
*Spondias Mombin*	-	-	-	-	-	-	-	-
**Urticaceae**								
*Cecropia* sp.	1.9	22.6	20.3	11.1	4.8	0.3	10.9	20.4
**Myrtaceae**								
Myrtaceae sp.	0.2	1.8	1.1	0.9	-	0.5	0.9	0.4
**Solanaceae**								
*Solanum* sp.	-	10.7	5.9	8.2	1.3	-	0.7	0.8
**Burseraceae**								
*Protium* sp.	-	-	29.9	45.0	1.1	-	0.4	0.8
**Piperaceae**								
*Piper marginatum*	0.1			0.3	0.2			
**Fabaceae**								
*Diplotropis purpurea*	0.1	-	-	0.6	-	-	-	-
**Asteraceae**								
Asteraceae sp.	-	-	0.7	-	-	-	-	-
**Icacinaceae**								
*Emmotum fagifolium*	-	-	-	-	-	-	-	0.8
**Chenopodiaceae/Amaranthaceae**								
Chenopodiaceae/Amaranthaceae	-	-	-	0.6	1.8	-	3.8	12.2
**Scrophulariaceae**								
Scrophulariaceae sp.	-	-	-	-	2.4	-	-	-
PG/10 g	289,000	38,400	42,588	7809	11,510	118,400	14,642	10,920
Honey class (I to V)	III	II	II	I	I	II	I	I
Honeydew indicators	-	-	R	R	R	R	VF	R
Electrical conductivity (mS/cm)	0.72	0.51	0.75	1.02	0.91	0.75	0.59	0.65
Color Pfund (mm)	99	62	51	62	51	71	35	41
Moisture (%)	18.3	18.7	18.1	17.9	18.5	17.8	18.9	19.8

RF > 45%: dominant pollen forms; RF = 16–45%: secondary pollen forms; RF = 3–16%: important minor pollen; RF < 3%: minor pollen. Capital letters represent the frequency of honeydew indicators: R: rare; F: frequent; VF: very frequent. According to the total number of pollen grains, honey is placed into one of the following 5 classes: Class I (PG/10 g ≤ 20,000); Class II: (20,000 < PG/10 g ≤ 100,000); Class III (100,000 < PG/10 g ≤ 500,000); Class IV (500,000 < PG/10 g ≤ 1,000,000); Class V (PG/10 g > 1,000,000).

**Table 4 foods-13-01073-t004:** Pollen spectra of honey collected in Saint-Laurent-du-Maroni from French Guiana.

**Pollen Type** Present in more than 50% of samples with RF at least once greater than 10% found in the honey samples from French Guiana (across all sampling sites)	**Saint-Laurent-Du-Maroni**									
	**Site 2**	**Site 3**								
	H45	H1	H2	H7	H13	H33	H46	H83	H84	H67
	Nov, 2015	Sept, 2014	sept, 2014	Nov, 2014	Jan, 2015	Sept, 2015	Dec, 2015	Nov, 2016	Nov, 2016	Mar, 2017
**Mimosaceae**										
*Mimosa pudica*	18.1	96.6	97.7	54.0	24.5	97.8	14.2	85.0	59.2	56.3
**Arecaceae**										
*Cocos* sp.	1.0	-	-	0.1	9.2	0.1	2.5	0.1	2.7	1.3
*Mauritia flexuosa*	1.3	-	-	0.4	-	-	3.5	1.4	2.6	2.6
**Cyperaceae**										
*Rhynchospora* sp.	1.0	0.1	0.2	5.9	0.2	0.1	-	-	1.1	0.8
**Acanthaceae**										
*Avicennia germinans*	1.0	-	-	-	-	-	0.2	0.1	0.7	2.0
**Anacardiaceae**										
*Tapirira guianensis*	1.0	1.1	0.3	17.8	8.7	0.1	56.8	6.5	22.3	14.4
*Spondias Mombin*	-	-	-	-	-	-	0.3	0.2	0.3	0.8
**Urticaceae**										
*Cecropia* sp.	35.2	0.1	0.5	3.1	16.8	0.3	3.5	1.7	3.0	13.5
**Myrtaceae**										
Myrtaceae sp.	0.3	-	-	0.2	-	-	-	0.2	0.3	0.5
**Solanaceae**										
*Solanum* sp.	0.9	-	0.1	0.1	-	-	-	0.5	0.3	-
**Burseraceae**										
*Protium* sp.	1.0	-	-	0.7	-	-	2.5	-	-	-
**Piperaceae**										
*Piper marginatum*		-	0.1	-	-	-	-	-	-	-
**Fabaceae**										
*Diplotropis purpurea*	-	0.9	-	14.5	15.1	-	0.5	0.2	1.1	1.1
**Asteraceae**										
Asteraceae sp.	-	-	-	0.2	10.0	-	-	0.1	0.3	0.2
**Icacinaceae**										
*Emmotum fagifolium*	0.1	-	-	-	-	-	-	-	-	-
**Chenopodiaceae/Amaranthaceae**										
Chenopodiaceae/Amaranthaceae	0.3	-	-	-	8.3	-	-	0.1	1.0	1.1
**Scrophulariaceae**										
Scrophulariaceae sp.	33.8	-	-	-	-	-	10.8	-	-	-
PG/10 g	31,700	52,4300	476,400	18,450	41,900	403,000	38,500	88,305	47,351	51,980
Honey class (I to V)	II	IV	III	I	II	III	II	II	II	II
Honeydew indicators	-	-	R	F	-	-	R	-	F	VF
electrical conductivity (mS/cm)	0.74	0.48	0.83	0.93	0.82	0.60	0.74	0.92	0.79	0.92
Color Pfund (mm)	71	62	83	55	41	83	55	83	71	99
Moisture (%)	18.1	21.4	21.9	20	20.5	18.5	17.8	20.1	19.4	18.5

RF > 45%: dominant pollen forms; RF = 16–45%: secondary pollen forms; RF = 3–16%: important minor pollen; RF < 3%: minor pollen. Capital letters represent the frequency of honeydew indicators: R: rare; F: frequent; VF: very frequent. According to the total number of pollen grains, honey is placed into one of the following 5 classes: Class I (PG/10 g ≤ 20,000); Class II: (20,000 < PG/10 g ≤ 100,000); Class III (100,000 < PG/10 g ≤ 500,000); Class IV (500,000 < PG/10 g ≤ 1,000,000); Class V (PG/10 g > 1,000,000).

**Table 5 foods-13-01073-t005:** Pollen spectra of honey collected in Sinnamary (seaside honeys) from French Guiana.

**Pollen Type** Present in more than 50% of samples with RF at least once greater than 10% found in the honey samples from French Guiana (across all sampling sites)	**Sinnamary (Seaside Honeys)**																		
	**Sites 4 and 5**																		
	H3	H8	H14	H18	H19	H20	H22	H24	H25	H54	H56	H58	H60	H61	H74	H76	H78	H80	H81
	Aug, 2014	Nov, 2014	Jan, 2015	Jul, 2015	Jul, 2015	Aug, 2015	Sept, 2015	Sept, 2015	Sept, 2015	Oct, 2015	Nov, 2015	Nov, 2015	Oct, 2015	Nov, 2015	Oct, 2016	Oct, 2016	Nov, 2016	Dec, 2016	Dec, 2016
**Mimosaceae**																			
*Mimosa pudica*	96.4	80.3	33.9	96.6	98.7	96.1	96.0	93.8	93.6	76.3	55.2	41.0	88.4	14.7	78.8	79.5	31.1	40.7	22.6
**Arecaceae**																			
*Cocos* sp.	-	0.7	10.8	0.2	0.1	0.2	0.3	1.0	0.7	2.9	4.1	4.0	1.8	5.5	3.0	1.5	2.9	6.0	7.2
*Mauritia flexuosa*	-	1.8	2.0	-	-	0.1	-	0.1	0.1	0.4	3.4	5.7	-	3.0	1.6	2.3	11.8	2.5	0.6
**Cyperaceae**																			
*Rhynchospora* sp.	0.1	0.5	-	-	-	0.1	0.1	0.2	0.3	0.6	1.7	0.7	0.2	-	0.5	0.6	0.2	2.3	5.3
**Acanthaceae**																			
*Avicennia germinans*	1.0	6.7	14.1	0.4	0.1	0.3	0.6	1.2	1.3	6.3	21.6	23.1	3.5	17.7	8.8	10.0	44.0	29.1	41.6
**Anacardiaceae**																			
*Tapirira guianensis*	-	-	4.8	-	-	-	-	0.1	0.4	-	-	14.8	-	38.1	2.3	1.9	2.4	5.3	4.7
*Spondias Mombin*	-	-	0.4	-	-	-	-	-	-	0.1	0.2	5.6	-	10.9	-	-	0.2	0.7	4.7
**Urticaceae**																			
*Cecropia* sp.	0.1	-	-	0.5	0.2	-	0.2	0.4	0.3	0.9	0.3	0.3	1.1	0.4	0.4	0.2	0.3	0.4	1.3
**Myrtaceae**																			
Myrtaceae sp.	-	0.7	14.3	-	-	-	0.3	0.2	0.2	0.4	1.1	0.7	0.2	3.2	0.4	1.0	0.3	0.2	-
**Solanaceae**																			
*Solanum* sp.	0.1	3.4	2.2	0.2	0.1	-	0.3	1.4	0.8	1.2	3.8	0.7	1.4	0.4	0.4	1.0	2.0	0.9	0.2
**Burseraceae**																			
*Protium* sp.	0.2	4.1	-	-	-	-	-	0.1	0.0	8.2	3.6	0.9	1.3	0.6	0.2	0.1	-	0.8	0.2
**Piperaceae**																			
*Piper marginatum*								0.1	0.1	0.1	0.6	0.6	-	-	-	0.1	-	0.2	0.9
**Fabaceae**																			
*Diplotropis purpurea*	0.1	-	0.2	-	-	1.3	0.6	0.3	0.3	0.1	-	-	-	-	1.3	0.0	1.0	0.1	0.2
**Asteraceae**																			
Asteraceae sp.	0.1	-	-	-	-	0.01	0.04	-	-	-	-	-	-	-	-	-	-	-	-
**Icacinaceae**																			
*Emmotum fagifolium*					0.01					0.2			0.3			0.0		0.1	
**Chenopodiaceae/Amaranthaceae**																			
Chenopodiaceae/Amaranthaceae	-	-	-	-	-	-	-	-	-	-	-	-	-	-	-	-	-	-	-
**Scrophulariaceae**																			
Scrophulariaceae sp.	-	-	-	-	-	-	-	-	-	-	-	-	-	-	-	-	-	-	-
PG/10 g	317,840	19,892	22,800	540,000	637,500	452,000	368,000	263,300	256,200	63,100	29,208	63,800	124,700	40,250	97,613	126,811	38,617	36,091	15,413
Honey class (I to V)	III	I	II	IV	IV	III	III	III	III	II	II	II	III	II	II	III	II	II	I
Honeydew indicators	R	R	R	R	-	-	-	-	-	R	R	R	R	R	-	-	-	F	F
Electrical conductivity (mS/cm)	0.55	0.64	0.59	0.77	0.77	0.64	0.87	0.93	0.93	0.62	0.49	0.50	0.65	0.49	0.46	0.46	0.44	0.46	0.44
Color Pfund (mm)	41	41	35	62	83	71	71	51	55	35	27	27	62	35	71	55	41	41	35
Moisture (%)	18.7	16.7	18.4	17.6	17.2	17.3	17	16.6	17.3	16.7	16.8	16.8	16.8	16.8	18.2	18.3	18.8	19.1	19.1

RF > 45%: dominant pollen forms; RF = 16–45%: secondary pollen forms; RF = 3–16%: important minor pollen; RF < 3%: minor pollen. Capital letters represent the frequency of honeydew indicators: R: rare; F: frequent; VF: very frequent. According to the total number of pollen grains, honey is placed into one of the following 5 classes: Class I (PG/10 g ≤ 20,000); Class II: (20,000 < PG/10 g ≤ 100,000); Class III (100,000 < PG/10 g ≤ 500,000); Class IV (500,000 < PG/10 g ≤ 1,000,000); Class V (PG/10 g > 1,000,000).

**Table 6 foods-13-01073-t006:** Pollen spectra of honey collected in Sinnamary (Amazonian forest honeys) from French Guiana.

**Pollen Type** Present in more than 50% of samples with RF at least once greater than 10% found in the honey samples from French Guiana (across all sampling sites)	**Sinnamary (Amazonian Forest Honeys)**																		
	**Sites 6 and 7**																		
	H4	H9	H15	H21	H23	H26	H27	H52	H53	H55	H57	H69	H70	H71	H72	H75	H77	H79	H82
	Aug, 2014	Nov, 2014	Jan, 2015	Aug, 2015	Sept, 2015	Sept, 2015	Oct, 2015	Sept, 2015	Oct, 2015	Oct, 2015	Nov, 2015	Sept, 2016	Oct, 2016	Sept,2016	Oct, 2016	Oct, 2016	Oct, 2016	Nov, 2016	Dec, 2016
**Mimosaceae**																			
*Mimosa pudica*	97.9	32.8	51.9	92.5	96.9	94.9	67.1	75.7	94.3	94.5	68.9	96.7	96.1	94.0	89.2	86.9	84.1	32.3	35.9
**Arecaceae**																			
*Cocos* sp.	-	6.6	1.2	0.1	0.1	0.5	3.4	3.8	0.2	1.0	2.1	0.3	0.6	0.8	1.0	1.8	0.7	2.7	4.6
*Mauritia flexuosa*	-	16.7	7.0	0.2	-	-	-	0.4	-	0.5	7.0	-	-	-	-	0.7	2.9	9.2	1.2
**Cyperaceae**																			
*Rhynchospora* sp.	-	0.6	0.7	0.1	-	0.5	0.1	1.0	-	0.1	0.6	0.0	0.2	0.3	0.7	1.2	1.0	15.1	3.1
**Acanthaceae**																			
*Avicennia germinans*	0.2	7.6	5.0	0.1	-	0.5	7.1	6.1	1.1	1.1	9.6	0.1	-	0.7	1.9	2.1	3.9	15.3	30.7
**Anacardiaceae**																			
*Tapirira guianensis*	-	2.1	3.9	0.1	-	-	-	0.1	-	0.1	0.3	0.1	0.1	0.2	2.7	3.1	2.1	0.8	2.7
*Spondias Mombin*	-	0.2	-	-	-	-	-	0.1	-	-	-	0.1	-	-	-	-	-	0.1	2.7
**Urticaceae**																			
*Cecropia* sp.	-	-	-	0.1	-	0.2	0.1	0.5	-	-	0.1	0.2	0.2	0.2	0.3	0.1	-	-	-
**Myrtaceae**																			
Myrtaceae sp.	-	2.9	9.5	0.1	-	-	-	-	0.2	0.1	0.2	-	-	-	-	0.5	1.2	0.9	1.7
**Solanaceae**																			
*Solanum* sp.	-	-	-	-	-	0.1	1.0	1.3	0.6	0.2	-	0.1	0.3	-	0.1	0.3	0.1	0.2	-
**Burseraceae**																			
*Protium* sp.	-	6.1	0.2	-	-	-	0.6	8.5	0.0	0.1	0.9	0.1	-	-	0.1	-	0.1	0.4	0.6
**Piperaceae**																			
*Piper marginatum*	-	-	-	-	0.1	-	-	0.3	-	-	-	0.04	0.4	1.0	0.9	0.3	-	0.1	-
**Fabaceae**																			
*Diplotropis purpurea*	-	-	-	0.3	0.1	-	-	-	0.1	0.1	0.3	0.1	-	-	-	0.6	0.3	-	-
**Asteraceae**																			
Asteraceae sp.	-	-	-	-	-	-	-	0.1	-	-	-	-	-	-	-	-	-	0.1	-
**Icacinaceae**																			
*Emmotum fagifolium*	-	-	-	-	-	-	15.4	0.2	0.9	0.1	0.5	-	-	0.1	-	-	-	1.9	0.2
**Chenopodiaceae/Amaranthaceae**																			
Chenopodiaceae /Amaranthaceae	-	-	-	-	-	-	-	-	-	-	-	-	-	-	-	-	-	-	-
**Scrophulariaceae**																			
Scrophulariaceae sp.	-	-	-	-	-	-	-	-	-	-	-	-	-	-	-	-	-	-	-
PG/10 g	1,971,334	44,674	37,320	880,800	475,000	420,000	38,600	62,600	71,800	510,250	77,600	488,063	548,332	439,770	235,555	205,330	101,038	109,772	22,691
Honey class (I to V)	V	II	II	IV	III	III	II	II	II	IV	II	III	IV	III	III	III	III	III	II
Honeydew indicators	R	R	-	R	R	R	R	R	R	R	R	R	R	F	R	R	F	F	F
Electrical conductivity (mS/cm)	0.55	0.45	0.54	0.63	0.86	0.96	0.74	0.62	0.71	0.65	0.57	0.55	0.35	0.44	0.36	0.43	0.48	0.53	0.51
Color Pfund (mm)	83	46	41	92	83	55	27	35	62	83	51	110	110	110	92	92	72	62	35
Moisture (%)	17.1	17.9	18.1	17	17.1	17.6	16.5	16.7	17.1	17	16.4	18.6	19.2	18.7	19.1	18.3	18.2	18.4	19.3

RF > 45%: dominant pollen forms; RF = 16–45%: secondary pollen forms; RF = 3–16%: important minor pollen; RF < 3%: minor pollen. Capital letters represent the frequency of honeydew indicators: R: rare; F: frequent; VF: very frequent. According to the total number of pollen grains, honey is placed into one of the following 5 classes: Class I (PG/10 g ≤ 20,000); Class II: (20,000 < PG/10 g ≤ 100,000); Class III (100,000 < PG/10 g ≤ 500,000); Class IV (500,000 < PG/10 g ≤ 1,000,000); Class V (PG/10 g > 1,000,000).

**Table 7 foods-13-01073-t007:** Pollen spectra of honey collected in Macouria/Kourou from French Guiana.

**Pollen Type** Present in more than 50% of samples with RF at least once greater than 10% found in the honey samples from French Guiana (across all sampling sites)	**Macouria/Kourou**						
	**Sites 8, 9 and 10**				**Site 11**		
	H37	H36	H38	H73	H42	H43	H44
	Dec, 2014	Sept, 2015	Nov, 2015	Oct, 2016	Oct, 2015	Nov, 2015	Dec, 2015
**Mimosaceae**							
*Mimosa pudica*	20.0	93.3	67.5	78.5	88.8	81.2	9.4
**Arecaceae**							
*Cocos* sp.	2.2	0.6	1.0	0.9	1.0	0.6	0.4
*Mauritia flexuosa*	1.6	0.0	0.8	1.3	0.2	12.9	0.1
**Cyperaceae**							
*Rhynchospora* sp.	0.1	0.1	0.1	0.2	0.2	1.7	0.4
**Acanthaceae**							
*Avicennia germinans*	6.7	0.3	7.3	3.1	5.2	0.4	0.7
**Anacardiaceae**							
*Tapirira guianensis*	50.1	0.4	-	9.6	0.1	0.2	83.4
*Spondias Mombin*	-	-	0.2	0.1	-	-	-
**Urticaceae**							
*Cecropia* sp.	-	0.2	0.2	-	2.2	-	-
**Myrtaceae**							
Myrtaceae sp.	1.7	0.4	0.2	0.9	-	-	-
**Solanaceae**							
*Solanum* sp.	14.8	2.4	18.5	0.7	-	-	-
**Burseraceae**							
*Protium* sp.	0.9	-	1.3	-	-	-	-
**Piperaceae**							
*Piper marginatum*	-	0.2	-	0.1	-	-	-
**Fabaceae**							
*Diplotropis purpurea*	-	0.9	-	0.4	-	-	-
**Asteraceae**							
Asteraceae sp.	-	-	-	0.1	0.1	-	-
**Icacinaceae**							
*Emmotum fagifolium*	-	-	0.2	-	-	-	-
**Chenopodiaceae/Amaranthaceae**							
Chenopodiaceae/Amaranthaceae	-	-	-	-	-	-	-
**Scrophulariaceae**							
Scrophulariaceae sp.	0.1	-	-	-	-	-	-
PG/10 g	78,364	192,000	47,950	141,288	91,030	555,464	1,176,118
Honey class (I to V)	II	III	II	III	II	IV	V
Honeydew indicators		R	R	-	R	-	-
Electrical conductivity (mS/cm)	0.62	0.88	0.70	0.64	0.47	0.79	0.61
Color Pfund (mm)	71	83	46	92	27	83	55
Moisture (%)	19.2	18.5	18.2	18.3	18.3	21.4	20.5

RF > 45%: dominant pollen forms; RF = 16–45%: secondary pollen forms; RF = 3–16%: important minor pollen; RF < 3%: minor pollen. Capital letters represent the frequency of honeydew indicators: R: rare; F: frequent; VF: very frequent. According to the total number of pollen grains, honey is placed into one of the following 5 classes: Class I (PG/10 g ≤ 20,000); Class II: (20,000 < PG/10 g ≤ 100,000); Class III (100,000 < PG/10 g ≤ 500,000); Class IV (500,000 < PG/10 g ≤ 1,000,000); Class V (PG/10 g > 1,000,000).

**Table 8 foods-13-01073-t008:** Pollen spectra of honey collected in Montsinéry-Tonnégrande from French Guiana.

**Pollen Type** Present in more than 50% of samples with RF at least once greater than 10% found in the honey samples from French Guiana (across all sampling sites)	**Montsinéry-Tonnégrande**																			
	**Site 12**	**Site 13**							**Site 14**											**Site 15**
	H30	H6	H11	H12	H16	H17	H29	H31	H39	H40	H50	H51	H59	H63	H64	H65	H66	H86	H87	H28
	Sept, 2015	Sept, 2014	Nov, 2014	Nov, 2014	Dec, 2014	Dec, 2014	Sept, 2015	Oct, 2015	Sept, 2015	Oct, 2015	Nov, 2015	Dec, 2015	Jan, 2016	Sept, 2016	Oct, 2016	Oct, 2016	Nov, 2016	Dec, 2016	Jan, 2017	Sept, 2015
**Mimosaceae**																				
*Mimosa pudica*	94.4	98.6	34.7	98.1	47.1	82.3	94.9	91.2	98.6	93.1	73.3	58.3	54.1	94.9	91.0	87.6	91.2	95.4	85.4	97.2
**Arecaceae**																				
*Cocos* sp.	0.6	0.3	30.5	0.4	10.5	3.2	0.4	1.9	0.1	0.4	3.9	3.0	6.8	0.1	0.3	0.3	0.1	0.5	0.8	0.2
*Mauritia flexuosa*	-	-	12.2	-	0.1	2.1	-	-	-	-	0.8	1.5	0.7	-	0.1	1.3	1.3	0.5	0.6	-
**Cyperaceae**																				
*Rhynchospora* sp.	0.3	0.1	2.0	0.2	4.4	0.6	0.3	1.2	0.4	1.0	1.0	3.3	3.9	0.1	0.1	0.3	0.5	0.3	1.6	0.2
**Acanthaceae**																				
*Avicennia germinans*	1.1	0.4	4.4	0.1	0.5	1.1	0.5	1.1	-	-	1.4	1.8	2.7	-	0.1	1.4	0.9	1.0	0.6	-
**Anacardiaceae**																				
*Tapirira guianensis*	0.2	-	0.5	-	20.3	1.2	1.0	0.6	0.2	0.1	-	14.8	10.9	3.4	6.9	4.9	2.8	0.3	4.2	-
*Spondias Mombin*	-	0.1	6.5	-	-	0.3	-	-	-	0.1	7.9	4.2	11.1	-	-	-	0.1	0.6	-	-
**Urticaceae**																				
*Cecropia* sp.	0.4	-	0.6	0.3	0.1	0.3	1.3	0.8	-	0.4	0.2	-	-	-	0.1	1.8	0.8	0.5	0.6	1.0
**Myrtaceae**																				
Myrtaceae sp.	0.4	0.1	0.8	0.3	10.7	0.4	0.5	1.6	-	0.2	0.9	0.6	0.4	-	0.3	0.9	0.3	0.3	0.4	-
**Solanaceae**																				
*Solanum* sp.	0.2	0.02	0.1	-	0.1	0.6	0.1	-	-	-	-	0.1	-	-	-	-	-	0.1	-	0.9
**Burseraceae**																				
*Protium* sp.	0.1	-	6.0	0.1	0.1	5.4	-	0.2	-	-	0.1	0.9	0.1	-	-	1.0	0.7	0.2	0.6	-
**Piperaceae**																				
*Piper marginatum*	-	-	-	-	0.04	0.1	-	-	0.1	-	-	-	-	-	-	-	-	-	-	-
**Fabaceae**																				
*Diplotropis purpurea*	0.1	-	-	-	-	0.3	-	-	-	-	-	-	0.2	-	-	-	-	-	-	-
**Asteraceae**																				
Asteraceae sp.	0.3	-	-	0.02	0.04	0.06	0.1	-	-	0.1	0.1	0.2	0.3	0.02	-	-	0.1	-	0.1	-
**Icacinaceae**																				
*Emmotum fagifolium*	0.4	-	-	-	-	-	0.1	0.04	-	3.8	1.9	0.1	0.7	0.02	-	-	-	-	-	-
**Chenopodiaceae/Amaranthaceae**																				
Chenopodiaceae/Amaranthaceae	-	-	-	-	-	-	-	-	-	-	-	-	-	-	-	-	-	-	-	-
**Scrophulariaceae**																				
Scrophulariaceae sp.	-	-	-	-	-	-	-	-	-	-	-	-	-	-	-	-	-	-	-	-
PG/10 g	109,000	252,800	44,450	502,000	126,700	81,104	294,000	182,500	300,200	278,300	118,400	43,840	92,120	447,650	455,300	93,332	127,070	63,877	93,845	218,400
Honey class (I to V)	III	III	II	IV	III	II	III	III	III	III	III	II	II	III	III	II	III	II	II	III
Honeydew indicators	-	R	-	-	R	F	R	R	F	R	R	R	R	R	R	-	R	-	-	R
Electrical conductivity (mS/cm)	0.71	0.67	0.77	0.52	1.04	0.63	0.76	0.65	1.09	1.04	0.99	0.83	0.85	0.79	0.80	0.98	0.87	0.95	0.99	1.22
Color Pfund (mm)	55	41	51	83	55	55	71	46	55	51	55	51	62	62	51	46	51	35	62	62
Moisture (%)	18.1	18.4	20.1	19.4	20.7	18.5	17.6	18.4	16.7	17.5	17.3	16.4	15.8	21	19	19.7	17.9	18.4	18.7	19.0

RF > 45%: dominant pollen forms; RF = 16–45%: secondary pollen forms; RF = 3–16%: important minor pollen; RF < 3%: minor pollen. Capital letters represent the frequency of honeydew indicators: R: rare; F: frequent; VF: very frequent. According to the total number of pollen grains, honey is placed into one of the following 5 classes: Class I (PG/10 g ≤ 20,000); Class II: (20,000 < PG/10 g ≤ 100,000); Class III (100,000 < PG/10 g ≤ 500,000); Class IV (500,000 < PG/10 g ≤ 1,000,000); Class V (PG/10 g > 1,000,000).

**Table 9 foods-13-01073-t009:** Pollen spectra of honey collected in Rémire-Montjoly from French Guiana.

**Pollen Type** Present in more than 50% of samples with RF at least once greater than 10% found in the honey samples from French Guiana (across all sampling sites)	**Rémire-Montjoly**			
	**Site 16**			
	H5	H10	H41	H68
	Nov, 2012	Nov, 2014	Nov, 2015	Nov, 2016
**Mimosaceae**				
*Mimosa pudica*	34.4	48.4	56.6	40.3
**Arecaceae**				
*Cocos* sp.	2.9	0.4	0.9	4.2
*Mauritia flexuosa*	0.7	-	-	-
**Cyperaceae**				
*Rhynchospora* sp.	1.4	3.3	0.3	3.4
**Acanthaceae**				
*Avicennia germinans*	5.0	1.1	1.7	2.9
**Anacardiaceae**				
*Tapirira guianensis*	1.6	13.6	8.1	19.6
*Spondias Mombin*	34.2	15.4	27.8	21.6
**Urticaceae**				
*Cecropia* sp.	-	-	0.2	0.5
**Myrtaceae**				
Myrtaceae sp.	0.9	0.2	0.1	0.3
**Solanaceae**				
*Solanum* sp.	-	-	-	-
**Burseraceae**				
*Protium* sp.	-	-	-	-
**Piperaceae**				
*Piper marginatum*	4.5	14.7	1.6	0.5
**Fabaceae**				
*Diplotropis purpurea*	-	0.2	-	-
**Asteraceae**				
Asteraceae sp.	0.6	0.5	-	0.5
**Icacinaceae**				
*Emmotum fagifolium*	-	-	-	-
**Chenopodiaceae/Amaranthaceae**				
Chenopodiaceae/Amaranthaceae	-	-	-	-
**Scrophulariaceae**				
Scrophulariaceae sp.	-	-	-	-
PG/10 g	71,384	48,720	36,634	60,763
Honey class (I to V)	II	II	II	II
Honeydew indicators	R	F	F	-
Electrical conductivity (mS/cm)	0.58	0.72	0.77	0.63
Color Pfund (mm)	83	92	46	55
Moisture (%)	17.1	18.6	18.5	20.7

RF > 45%: dominant pollen forms; RF = 16–45%: secondary pollen forms; RF = 3–16%: important minor pollen; RF < 3%: minor pollen. Capital letters represent the frequency of honeydew indicators: R: rare; F: frequent; VF: very frequent. According to the total number of pollen grains, honey is placed into one of the following 5 classes: Class I (PG/10 g ≤ 20,000); Class II: (20,000 < PG/10 g ≤ 100,000); Class III (100,000 < PG/10 g ≤ 500,000); Class IV (500,000 < PG/10 g ≤ 1,000,000); Class V (PG/10 g > 1,000,000).

**Table 10 foods-13-01073-t010:** Statistical analysis of pollen profiles from different localities (pollen grains present in more than 50% of the samples and an RF at least one greater than 16%).

		**Awala**	**SLT**	**SIN1**	**SIN2**	**MAK**	**MTG**	**RMJ**
**Number Sample**		**8**	**10**	**19**	**19**	**7**	**8**	**4**
*Cocos* sp.	min.–max.	0.3–9.0	0.0–9.2	0.0–10.8	0.0–6.6	0.4–2.2	0.1–30.5	0.4–4.2
	Mean ± SD	5.4 ± 3.5	1.7 ± 2.8	2.8 ± 2.9	1.7 ± 1.8	0.9 ± 0.6	3.2 ± 7.0	2.1 ± 1.77
	%RSD	64.4	165.5	105.0	108.4	64.7	2.2	84.5
*Mauritia* *Flexuosa*	min.–max.	0.0–14.0	0.0–3.5	0.0–11.8	0.0–16.7	0.0–12.9	0.0–12.2	0.0–0.7
	Mean ± SD	3.4 ± 4.8	1.2 ± 1.3	1.9 ± 2.9	2.4 ± 4.5	2.4 ± 4.7	1.1 ± 2.7	0.2 ± 0.4
	%RSD	140.7	112.2	154.1	186.1	194.0	2.6	200.0
*Avicennia* *germinans*	min.–max.	0.0–16.0	0.0–2.0	0.1–44.0	0.0–30.7	0.3–7.3	0.0–4.4	1.1–5.0
	Mean ± SD	4.9 ± 5.9	0.4 ± 0.7	**12.2 ± 13.9**	**4.9 ± 7.5**	3.4 ± 3.0	1.0 ± 1.1	2.7 ± 1.7
	%RSD	120.4	167.6	114.4	152.8	89.7	1.1	63.9
*Tapirira* *guianensis*	min.–max.	0.0–70.8	0.1–56.8	0.0–38.1	0.0–3.9	0.0–83.4	0.0–20.3	1.6–19.6
	Mean ± SD	**23.0 ± 30.1**	**12.9 ± 17.3**	3.9 ± 9.0	1.0 ± 1.3	**20.5 ± 33.2**	3.6 ± 5.6	10.7 ± 7.7
	%RSD	130.8	134.2	229.0	138.7	161.7	1.6	71.6
*Spondias* *mombin*	min.–max.	-	0.0–0.8	0.0–10.9	0.0–2.7	0.0–0.2	0.0–11.1	15.4–34.2
	Mean ± SD	-	0.2 ± 0.3	1.2 ± 2.8	0.2 ± 0.6	0.0 ± 0.1	1.5 ± 3.2	**24.7 ± 8.1**
	%RSD	-	167.6	238.0	368.8	174.3	2.1	32.6
*Cecropia* sp.	min.–max.	0.3–22.6	0.1–35.2	0.0–1.3	0.0–0.5	0.0–2.2	0.0–1.8	0.0–0.5
	Mean ± SD	**11.5 ± 8.8**	**7.8 ± 11.2**	0.4 ± 0.3	0.1 ± 0.1	0.4 ± 0.8	0.5 ± 0.5	0.2 ± 0.2
	%RSD	76.3	144.5	92.1	124.5	222.5	1.0	131.8
*Solanum* sp.	min.–max.	0.0–10.7	0.0–0.9	0.0–3.8	0.0–1.3	0.0–18.5	0.0–0.9	-
	Mean ± SD	3.4 ± 4.2	0.2 ± 0.3	1.1 ± 1.1	0.2 ± 0.4	**5.2 ± 7.9**	0.1 ± 0.2	-
	%RSD	122.5	158.6	100.3	160.9	153.0	2.1	-
*Protium* sp.	min.–max.	0.0–45.0	0.0–2.5	0.0–8.2	0.0–8.5	0.0–1.3	0.0–6.0	-
	Mean ± SD	**9.7 ± 17.6**	0.4 ± 0.8	1.1 ± 2.1	0.9 ± 2.3	0.3–0.6	0.8 ± 1.7	-
	%RSD	182.6	192.4	196.0	247.2	175.7	2.2	-
Scrophulariaceae sp.	min.–max.	0.0–2.4	0.0–33.8	-	-	0.0–0.1	-	-
	Mean ± SD	0.3 ± 0.8	**4.5 ± 10.8**	-	-	0.0 ± 0.0	-	-
	%RSD	282.8	243.5	-	-	264.6	-	-

## Data Availability

The original contributions presented in the study are included in the article and Appendix A, further inquiries can be directed to the corresponding author.

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
