# Peer review of "Melissopalynological Analysis of Honey from French Guiana"

_foods, 2024, doi:10.3390/foods13071073_

Round 1

Reviewer 1 Report

Comments and Suggestions for Authors

This manuscript deals with the assessment of the precise knowledge of plants foraged by honeybees. It gives significant attention to the melissopalynological analysis of honey as useful tool for understanding and determination of the botanical origin of honeys.

The manuscript is really good! It has tackled the majority of the most important empirical issues necessary for this topic. Noteworthy attention has been given to representative sampling that significantly elevates the overall quality of the paper. Manuscript has been well thought in its structure and presentation of the results. Good job!

Still, there are few minor/medium shortcomings for which the authors are suggested to address them with particular care.

1.      Authors should check their English. French words have been left and used instead.

2.      Authors should reconsider and better explain following statement:

Pollen grains content was calculated for each sample and constituted a key element for characterization to be considered. It showed that there is a link between density and pollen representation of species [10]. Thus, honeys issued from plant which have over-represented type of pollen, are always rich in pollen. Conversely, honeys from species of the under-represented type will always be very poor in pollen [10].

3.      Authors should have also insert the graphical presentation of pollen spectre (columns) in the manuscript. It may create some additional effort to authors; however, this has become the standard form of pollen presentation in papers and the authors should have put it in the main text.

4.      Authors should crosscheck the use of units throughout the text. Special emphasis is to be given to how are the units prescribed by Honey Directive or any other Honey Standard (i.e. IHC Standard Methods, CODEX Alimentarius etc.). If 0,9 mS/cm is prescribed as “honeydew honey limit”, why are they using the 900 µS/cm in the text? Yes, it is SI and mathematically correct, however in the light of standardization, as well as more comprehensible reading, authors are encouraged to perform the cross check throughout the manuscript text.  

Comments on the Quality of English Language

English should be given one more thorough check!

Reviewer 2 Report

Comments and Suggestions for Authors

There are Africanized honey bees in French Guiana, not Apis mellifera scutellata.

Check the abbreviation of genus names to make sure they are abbreviated after the first mention.

Figure 1: it is not possible to read the names on the maps, it is a blur. Suggestion: instead of listing the samples on the figure, list them on the legend or text. The important is to show the locations And by transferring them to the text, it is possible to increase the coordinates.

In section 2.2. Pollen sampling (fresh pollen): the correct is species not specie.

2.4. Honey slide’s pollen analysis and pollen density 

It is not clear if you counted one microscopic field or the entire slide at first. Then I saw that there is a number of total fields counted. How the NTC was chosen?

It is important to inform the magnification.

I suspect that the Physico-chemical test was done before the preparation of honeys for pollen analysis. It should be described before the sections on the palynological analysis and be specified when it was done.

Suggestion: include (RF) after 3.2 Relative frequency analysis to remind the reader what RF means.

There is no explanation about what relative presence analysis and relative frequency analysis in the material and methods. It should be included to make it clear when you are considering all honey samples or a mean per sample.

What do you mean by pollen spectra on section 3.2? Is it one sample?

3.4 French Guiana profile

The section name is missing an e on profile.

What is a normal and an overrepresented pollen type?

There is on M. pudica that is not in italics.

- I missed a figure for the results. Could it be done a neighbor joining analysis? Could it be included a bar graph showing the most frequent pollen types per region, for example?

Discussion:

“Vismia latifolia has an interesting geographic distribution”. What do you mean by interesting?

Melipona should be in italics.

How this great amount of pollen of Mimosa pudica ends up in honey if the species do not have nectaries?

One of the objectives of the study was to determine which species are nectariferous and polliniferous, but it is not clear in the results and discussion.

Simple summary: should it be a description of what it should be or is it missing?

Comments on the Quality of English Language

The English should be revised, minor adjustments are needed to meet formal parameters and improve the understanding of the text.

Reviewer 3 Report

Comments and Suggestions for Authors

The detail comments are given in the manuscript file.

The manuscript needs major revision. Abstract and Conclusion must be revised. The author must remove references from the results. Only one table is given in the manuscript whereas  all other tables are converted as supplementary data. It is suggested that author should redraft some part of the data from supplementary data and include it in the manuscript. 

Round 2

Reviewer 2 Report

Comments and Suggestions for Authors

Dear authors,

You have significantly improved the manuscript and incorporated all the required changes.

Congratulations on this important study.

Author Response

Thank you for your feedback. Your suggestions and comments have significantly improved our manuscript.

Best regards,

JIANG Weiwen.

Reviewer 3 Report

Comments and Suggestions for Authors

The author has significantly improved the manuscript according to the previous comments.

There are still certain minor corrections that are highlighted in the PDF file of attached manuscript. 

Author Response

Thank you for your feedback.

We have taken your comments and corrections into account (highlighted in grey color in the manuscript).

Best regards,

JIANG Weiwen.